# The Impact of Immune Checkpoint-Inhibitors Therapy in Urinary Bladder Cancer

Ana Lúcia Silva [1,†], Pedro Abreu-Mendes [2,3,4,†], Diana Martins [1,5,6,7] and Fernando Mendes [1,5,6,7,8,*]

1   Politécnico de Coimbra, ESTeSC, DCBL, Rua 5 de Outubro-SM Bispo, Apartado 7006,
    3046-854 Coimbra, Portugal; analsilva@estescoimbra.pt (A.L.S.); diana.martins@estescoimbra.pt (D.M.)
2   Urology Department, Centro Hospitalar e Universitário de São João, 4200-319 Porto, Portugal;
    up201903946@med.up.pt
3   Faculty of Medicine, University of Porto, 4200-319 Porto, Portugal
4   I3S—Instituto de Investigação e Inovação em Saúde, Universidade do Porto, 4200-135 Porto, Portugal
5   Faculty of Medicine, Coimbra Institute for Clinical and Biomedical Research (iCBR) Area of Environment
    Genetics and Oncobiology (CIMAGO), University of Coimbra, 3000-548 Coimbra, Portugal
6   Center for Innovative Biomedicine and Biotechnology (CIBB), University of Coimbra,
    3000-548 Coimbra, Portugal
7   Clinical Academic Center of Coimbra (CACC), 3000-548 Coimbra, Portugal
8   European Association for Professions in Biomedical Sciences, B-1000 Brussels, Belgium
*   Correspondence: fjmendes@estescoimbra.pt
†   Both authors contributed equally.

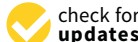



**Simple Summary:** Bladder cancer (BC) remains one of the most common and aggressive malignant diseases. Despite all the innovations, the effect on morbidity and mortality has been modest. Different immunotherapeutic treatments (namely bacillus Calmette–Guérin (BCG) intravesical instillation and anti-PD-1/PD-L1 immune checkpoint blockade) have been used in different stages of the disease, with some good results. For this reason, it is important to understand the association between inflammation, tumor microenvironment, and cancer to enlighten the further development of immunotherapy approaches. This review investigates new therapeutic approaches in urinary bladder cancer, namely, immune checkpoint inhibitors, to better understand the immune response and involved mechanisms.

**Abstract:** Bladder cancer (BC) is one of the most common cancers in the world. From an early age, it was observed that chronic inflammation is associated with conditions favorable to the development of tumors, as well as the tumor microenvironment. Moreover, regulating tumor progression also interferes with the therapy's response. The interaction between the tumor and the immune system led to the development of new immune therapies, the immune checkpoint inhibitors. Immunotherapy has shown a better safety profile, survival, and tolerance compared to standard chemotherapy. This therapy offers an effective alternative to patients who are ineligible for cisplatin and patients with advanced disease progression after platinum-based therapy. The first immunotherapy approved for BC was intravesical instillation with Bacillus Calmette–Guérin, for tumors at early stages. Later, immunotherapy focused on immune checkpoint inhibitors, namely, anti-programmed cell death protein 1 (PD1), anti-programmed cell death protein ligand 1(PD-L1), and anti-antigen 4 associated with cytotoxic T cells (CTLA-4). Currently, five immune checkpoint inhibitors for advanced BC are approved by the Food and Drug Administration (FDA): Atezolizumab, Durvalumab, Avelumab, Pembrolizumab, and Nivolumab. This review addresses the correlation between inflammation, tumor microenvironment, and cancer; various studies regarding immune checkpoint inhibitors, either in monotherapy or in combination therapy, are also addressed.

**Keywords:** BC; checkpoint inhibitors; immunotherapy; microenvironment

## 1. Introduction

The bladder is an organ located in the lower abdomen part, whose central role is to store urine, collected from the kidneys, via ureters, until urination [1]. Bladder cells are constantly exposed to numerous mutagens, which are filtered by the kidneys before reaching the bladder. Transitional epithelial cells covering the bladder, recognized as urothelial cells, accommodate the urine [1]. Bladder cancer (BC) consists of uncontrolled growth of neoplastic cells, initiated in the bladder. Its classification varies according to tumor histology [2,3].

In 2018, 549,393 new cases of BC were diagnosed worldwide, representing approximately 3% of all newly diagnosed cancer cases. In Europe, BC presented an incidence of 197,100 new cases and a total of 65, 000 deaths. The south of Europe presented the highest incidence, where 26.5/100,000 men and 5.5/100,000 women developed BC. Worldwide, men present a BC incidence four times higher than women, with a relative incidence of 9.6/100,000 and 2.4/100,000, respectively. Among men, BC is the sixth most incident cancer and the ninth most deadly [4,5].

Most BC arise secondarily from exogenous exposure to carcinogens through the gastrointestinal tract, respiratory tract, or skin contact [6]. Risk factors associated with the development of BC can be divided between hereditary (genetic predisposition) and acquired (environmental exposure) [7,8]. Among the environmental risk factors, smoking is the most common, being responsible for approximately 50% of BC. Smokers are 2.5 times more likely to develop BC than non-smokers. The second most common risk factor is exposure to carcinogens, namely, chlorinated hydrocarbons, aromatic amines, and polycyclic chromatic hydrocarbons. These substances are present in oil products, rubbers, dyes, inks, and metals [3,8]. Moreover, clinical therapies, namely, pelvic radiotherapy and cyclophosphamide therapy, have also been associated with BC development. Chronic inflammation is also a known established cause of BC [8].

Urothelial carcinoma represents approximately 90% of all cases of BC, being the most common histological subtype of BC [9,10]. Urothelial carcinoma is a heterogeneous pathology with multiple possible therapeutic approaches [11]. It is characterized by high prevalence and recurrence rates and seriously affecting the patient's quality of life [12,13]. From a pathological point of view, urothelial carcinoma presents two distinct subtypes: Non-muscle-invasive BC (NMIBC) and muscle-invasive BC (MIBC).

NMIBC is responsible for 75–80% of urothelial carcinoma cases. It is a subtype that generally does not represent a threat to the patient's survival. However, in most cases, it could present high recurrence rates, requiring lifelong surveillance [14]. Despite the high recurrence, it does not usually progress to MIBC [3]. MIBC is clinically more aggressive and often could lead to metastatic disease [14,15]. After the invasion of other organs, it is classified as incurable and has an average overall survival of 12 to 18 months [7]. BC patients are stratified by grade and tumor staging, according to the American Joint Committee on Cancer (AJCC) Staging Manual, as shown in Figure 1 [3].

The development of the tumor does not depend only on the autonomous characteristics of the cancer cells. It can also be correlated with factors, the immune microenvironment, immune system, and activation of signaling pathways (15). Thus, the therapeutic approaches to be applied depending on the location and avoidance of the tumor, staging, and physical condition of the patient (2). In the context of BC, intravesical instillation with BCG and immune checkpoint inhibitors are approved as therapies, including Atezolizumab, Pembrolizumab, Durvalumab, Nivolumab, and Avelumab. Currently, it has been concluded that urothelial carcinoma is an immunogenic tumor, which leads to immunotherapeutic approaches to demonstrate substantial clinical benefits [16].

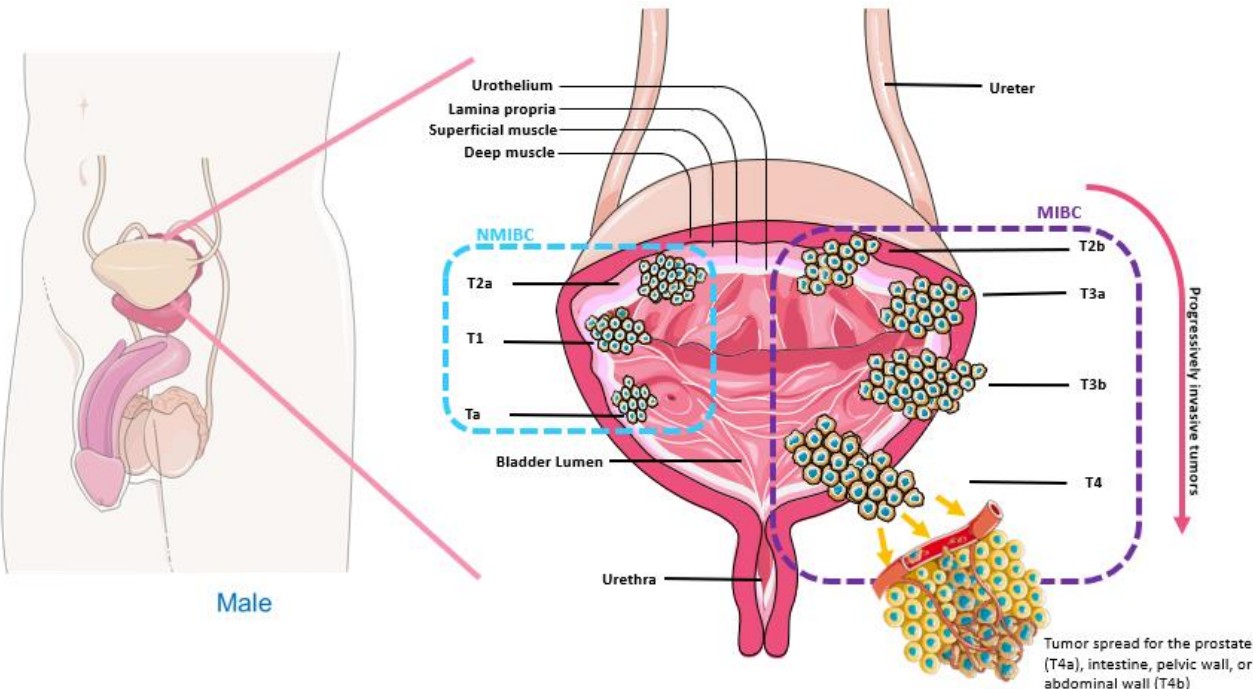

**Figure 1.** Bladder cancer (BC) staging system. The Tumor, Node, Metastases (TNM) system is applied to assign a standard extension of invasion (Ta–T4) and spread (N and M) [7,11,14]. The non-muscle-invasive BC (NMIBC) subtype (staging < T2) is limited to the mucous and submucosal inner lining of the bladder. On the other hand, the MIBC subtype (staging Table 2. T4) spreads to the detrusor lining of the bladder, with the possibility of wider spread, presenting a high risk of metastatic spread. T2 tumor is limited to the internal (T2a) or external (T2b) layer of the bladder wall. T3 tumor spreads beyond the muscular layer of the bladder wall, invading perivesical fat, microscopically (T3a) or macroscopically (T3b). T4 tumors directly invade the nearby organs, such as the prostate, uterus, or vagina (T4a), or intestine, pelvic wall, or abdominal wall (T4b).

## 1.1. Inflammation and Cancer

Inflammation is a physiological defense response against several events, such as pathogens, and ultimately promoting tissue repair [17,18]. Inflammation can be caused by physical damage, exposure to toxins, ischemic injury, infection, and other types of trauma [19]. If the trauma is maintained, the response is perpetuated and can evolve to a chronic inflammation state, resulting in cell mutation and proliferation, creating a favorable environment for cancer development [20]. Chronic inflammation has been associated with several stages of tumorigenesis, including cell proliferation and transformation, invasion, angiogenesis, and development of metastasis [19].

During the inflammation process, monocytes, neutrophils, and macrophages are recruited. This can lead to the production of reactive oxygen (ROS) and nitrogen (RNS) species. Consequently, there may occur damage of tissues, lipids, proteins, and DNA, leading to tumorigenesis and angiogenesis, as observed in BC [18,19].

The connection between inflammation and cancer can be intrinsic or extrinsic [21]. The intrinsic pathway is activated by events related to genetic factors, including activation of several oncogenes through mutation, chromosomal rearrangements or amplification, and inactivation of tumor suppressor genes. These genetic alterations lead to the activation and release of inflammatory mediators by the cell, which potentiates the inflammatory microenvironment. The extrinsic pathway represents the cell reaction when exposed to an external inflammation mediator—which also increases the potential for neoplasia. The main drivers of the intersection between these pathways include transcription factors, cytokines, such as interleukin 1 (IL-1), tumor necrosis factor (TNF), and chemokines. Thus, cancer-related inflammation is a key component of the tumor microenvironment [21].

Previous studies have shown that patients with BC have increased levels of various pro-inflammatory and anti-inflammatory cytokines, including IL-1, IL-4, IL-6, IL-8, IL-10, IL-17, IL-18, TNF-$\alpha$, and transforming growth factor-beta (TGF-$\beta$). This cellular heterogeneity is fundamental in the transition from inflammation to cancer [18,22].

Moreover, the crosstalk between tumor cells and the microenvironment can induce tumorigenesis and contribute to tumor progression. Initially, tumor cells and stromal cells release chemotactic factors leading to the recruitment of macrophages and neutrophils, among other cells [17]. The tumor itself can also induce damage to normal tissue and release damage-associated molecular patterns (DAMPs). These DAMPs, recognized by Toll-like receptors (TLRs), lead to the induction of innate immune responses producing inflammatory cytokines, interferon (IFN), and other mediators [17,23].

*1.2. Tumor Immune Microenvironment*

BC is a solid tumor that holds a complex system defined by the cellular components and tumor microenvironment that play an important role in tumor differentiation, immune escape, and metastases development [12,24]. It is a highly heterogeneous and dynamic environment consisting of tumor cells, immune infiltrating cells, stromal cells, vascular cells, and an extracellular matrix [25,26]. It presents a variable composition, regarding tumor microenvironment, recruiting tumor-associated macrophages (TAMs), myeloid-derived suppressor cells (MDSCs), tumor-infiltrating lymphocytes (TILs), and regulatory T cells (Tregs), creating an immunosuppressive environment that limits the intrusion of T effectors cells and their function, favoring the escape of immune surveillance by tumor cells [26,27].

Tumor-infiltrating lymphocytes are located in the vicinity of the tumor, being recruited to the tumor site to eliminate neoplasic cells. Their activity is regulated by tumor-specific immune mechanisms [12,26]. CD8+ T lymphocytes are a sub-population of cytotoxic lymphocytes which are responsible for removing target cells, including tumor cells. Studies suggest that CD8+ T lymphocytes infiltrated in tumors have anti-tumor functions, which could be responsible for favorable prognosis in several tumors [12,28]. An example of this is the stable or regressing metastases that often present infiltration by CD8+ and CD4+ T cells, while progressing metastases show a depletion of T cells [26].

Another type of cells that are also often recruited to the primary tumor and metastatic sites are the MDSCs, a population of immature myeloid, usually inhibiting the innate and adaptive immune reactions, by suppressing T cells (CD4 and CD8), as well as the Natural Killer cells (NK) [25]. Ornstein et al. study demonstrated the correlation between MDSCs with tumor staging, burden and response to therapy, and clinical outcomes [29].

TAMs participate directly in tumor development and progression [30]. Their origins are the blood monocytes or myeloid progenitors, that are recruited to the tumor. Once there, the tumor microenvironment promotes TAMs polarity, inducing them immunosuppressive, protecting tumor cells from immune system elimination.

TAMs are abundant in tumor stroma regardless of tumor staging. They secrete several factors that allow tumor growth through tumor invasion, angiogenesis inflammation, and immune escape. Besides, macrophages contribute to the development of metastases, creating a pre-metastatic niche, allowing extravasation and survival of tumor cells [25]. In vivo studies demonstrate a link between a higher frequency of infiltration of TAMs in tumors and a poor prognosis [30]. Correlation between tumor cells and tissue microenvironment establishes fundamental support for determining cancer severity. In this context, macrophages represent the main component of immune infiltrate that is often responsible for tumor rejection or progression, angiogenesis, and metastasis [31].

The tumor microenvironment is a heterogeneous entity among primary tumors. Within the same histological type, tumors present widely different tumor microenvironment. Cancer cells in the primary tumor are surrounded by an infiltrate formed by stromal fibroblasts, neutrophils, B lymphocytes, macrophages, T lymphocytes, and NK cells, causing immune cells to release soluble mediators to maintain an inflammatory microenvironment [21,26]. The analysis of tumor microenvironment components has a

great influence on prognosis. Observation of T cell infiltration is a prognostic indicator normally associated with a favorable prognosis. On the other hand, TAM infiltration is generally associated with a poor prognosis [21].

Immunity is fundamental in the tumor microenvironment pro-inflammatory cells, namely, TAMs, which may contribute to tumor progression simulating proliferation, angiogenesis also favoring a microenvironment advantageous to metastases development [18,21,22]. Other cells in the tumor microenvironment play a pro-metastatic role, such as immature myeloid cells or Tregs. Those cells suppress the immune response to growing tumors and release pro-angiogenic and growth factors, namely, tumor-associated fibroblasts and platelets, ultimately increasing the survival of circulating tumor cells [21].

Invasion and metastasis are essential for malignancy. Interaction between TAM and tumor-associated neutrophils with extracellular matrix shapes the tumor microenvironment and promotes tumor growth and spread. Quantification of the immune and inflammatory landscape of tumor microenvironment provided new prognostic indicators of cancer progression, as shown by quantification of infiltrated cells in tumors [21]. Interactions between tissue microenvironment and tumor cells are so important that they may determine the therapy response [27].

### 1.3. Immune Checkpoints

Immune checkpoints are fundamental in immune self-tolerance, preventing autoimmunity and shielding tissue damage, due to immune action [2,31]. On the other side, immune checkpoints are key regulators of the immune response, which negatively regulate T cells, allowing tumors to escape immune surveillance [2,32,33]. Tumors could present the ability to neutralize checkpoints to maintain immune resistance, sequestering them to restrict the ability of the immune system to develop a viable anti-tumor response [2].

The following immune checkpoints—programmed cell death protein 1 (PD-1), programmed cell death protein ligand 1 (PD-L1), as well as cytotoxic T-lymphocyte antigen 4 (CTLA-4) have been associated with BC [33,34]. PD-1 is a transmembrane protein of the immunoglobulin family, expressed in several immune cells, namely, on the surface of T lymphocytes, macrophages, and B lymphocytes [32,35]. So far, two PD-1 ligands, PD-L1 and PD-L2, have been identified [2,36]. These ligands are expressed in tumor cells, T and B cells, and epithelial cells. PD-1 acts as a checkpoint to prevent activation of T cells, which leads to decreased autoimmunity, allowing self-tolerance to be stimulated. Based on the above, the immune system is affected by the activity of PD-1, which suppresses, blocks, and inactivates immune cells, allowing cancer to grow, develop and progress [2].

PD-L1, PD-1 ligands, are a member of the immunoglobulin family. This protein is often expressed in several cells, namely, hematopoietic and non-hematopoietic cells, antigen-presenting cells, and also tumor cells [36–38]. High expression of PD-L1 has been shown to correlate with the advanced and aggressive staging of BC, with worse survival outcomes [25]. PD-1/PD-L1 interaction is used by cells to suppress T cell receptor-mediated cytotoxic function by inhibiting the proliferation of CD8+ T cells and reducing the production of gamma interferon (IFN-$\gamma$) and IL-2 [32,35]. This mechanism promotes self-tolerance, preventing self-attack by the immune system to its cells [33]. However, tumor cells also have these immune suppression mechanisms, either positively regulating the expression of PD-L1 or stimulating the expression of PD-L1 in tumor microenvironment cells [16,35].

CTLA-4 is a protein receptor molecule of the immunoglobulin family. It is expressed constitutively in Treg cells on the cell surface, but it can also be expressed by effector T cells after activation, mainly intracellularly [38,39]. Therefore, CTLA-4 expression by Treg cells acts as a suppressive mechanism to T cell responses.

On the other hand, intracellular CTLA-4 reservoirs prevent tissue damage caused by pathogenic auto-reactive T cells [40]. During T cell activation, CTLA-4 is regulated and competes against CD28 for CD80 (B7-1) and CD86 (B7-2) binding, inhibiting intracellular T cell activation and negatively regulating immune response [39]. CTLA-4 has played an important role in early immune response, especially in lymphoid tissues. On the other

hand, PD-1, whose expression is increased after activation of T cells in peripheral tissues, is related to a delayed immune response. Blocking these pathways (PD-1/PD-L1 and CTLA-4) by anti-PD-1, anti-PD-L1, or anti-CTLA-4 antibodies allows T cells to maintain their anti-tumor properties, and consequently, their ability to mediate tumor cell death [35,40,41].

Our aim, in this review, is to address the role of immune checkpoints in BC treatment.

## 2. Methods

The authors used the PRISMA (Preferred Reporting Items for Systematic Review and Meta-analysis) approach to perform a critical review.

Two of the authors conducted the literature search, acquiring the papers in an Embase/Pubmed search, related to randomized clinical trials (RCT), Scientific Article (SA), Review Articles (RA), and Clinical Trials (CT) with the following queries involving: "Bladder Cancer", "Checkpoint inhibitors", "Immunotherapy" and "Microenvironment".

All retrieved manuscripts were initially selected by title and abstract, in a total of 390 papers. Secondarily, an analysis was performed, considering the relevance and the quality of the previously selected papers for the theme, with a specific emphasis on manuscripts with novel findings regarding the above-mentioned therapies.

Papers were excluded when: (i) Not written in English, (ii) pre-clinical trials, (iii) papers supporting previously established data about these therapies, but out of date. Other exclusion criteria were established as unavailability of selected papers in free full text, papers published before 2017, studies in animals, and papers not written in English. Full details of the used search strategies are provided in Figure 2. Hence, after eliminating duplicated and nonrelevant articles, a total of 15 papers were included in this systematic update review from a total of 390 first identified.

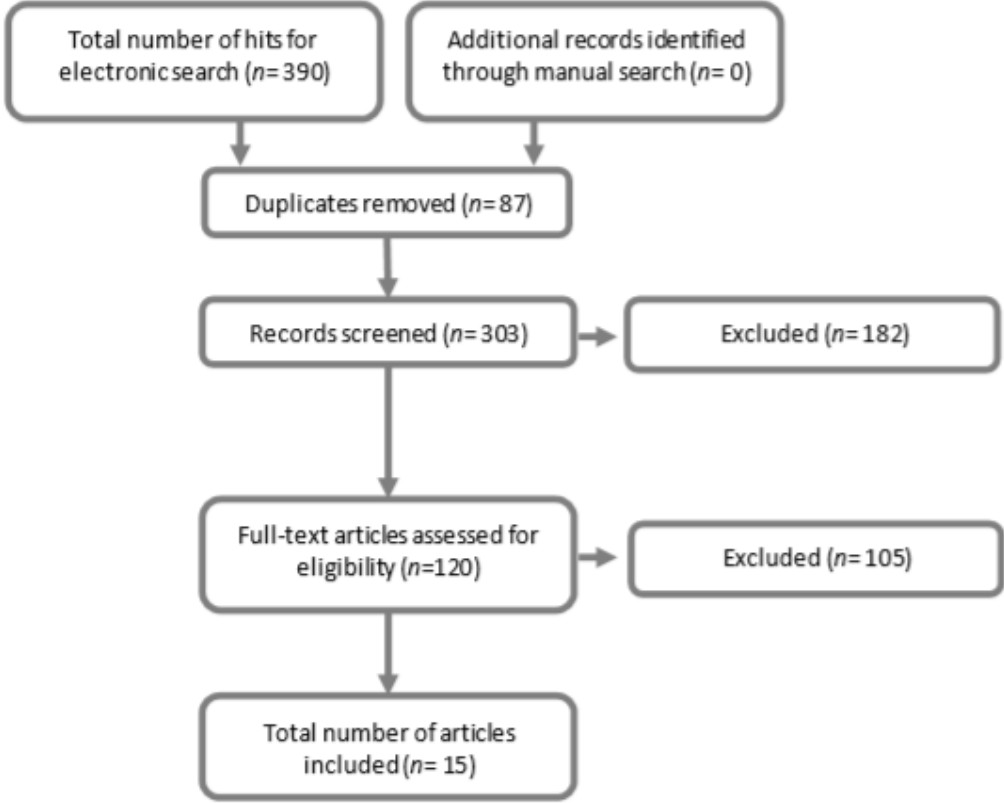

**Figure 2.** Flow chart. Prism of the systematic review, detailing the searches in the database, the number of abstracts selected, full-text articles assessed for eligibility, and the total number of articles included.

All ethical questions have been fulfilled, and all the sources that provided the theoretical support have been properly referenced.

### 3. Immunotherapy

Cancer immunotherapy has changed the paradigm of therapy for several types of cancer [42,43]. This therapeutic approach aims to improve anti-tumor immune response by reactivating the immune system that is silenced by tumor cells, through natural mechanisms [42,44]. Thus, immunotherapy is recognized as a promising strategy to treat certain types of cancer, presenting several advantages over non-specific therapies [43]. The main treatment methods used in cancer immunotherapy are cancer vaccines and monoclonal antibodies (mAb) [44].

#### 3.1. Bacillus Calmette-Guerin Intravesical Therapy

BCG is a non-pathogenic strain of *Mycobacterium bovis*. It was initially used as a vaccine for the prevention of Tuberculosis [2]. In 1976, its possible effectiveness in the prevention and treatment of NMIBC was documented for the first time [45,46]. In 1990, intravesical BCG in patients with BC was approved by the Food and Drug Administration (FDA) and became gold standard immunotherapy for urothelial carcinoma.

Currently, BCG is one of the most widely used immunotherapies in BC [25,47]. It's used in patients with intermediate or high-grade NMIBC (Ta, Tis, and T1) currently recommended [48]. BCG instillation in the bladder has been shown to develop a local and systemic innate immune response [2,25]. The bladder wall neutrophils and mononuclear cells' interaction with BC cells and BCG stimulates dendritic cell maturation. In turn, BCG-loaded dendritic cells will facilitate the immune inhibition of cancer cells [35]. The main function of this treatment is to stimulate, trigger and activate effector cells of the immune system to attack cancer cells by creating an influx of inflammatory cells, development of tissue macrophages resistance (M1), production of Th1 cytokines (IFN-$\gamma$, IL-12 e TNF-$\alpha$) and creation of Th1 mediated cytotoxic responses [2,27].

Despite initial success rates, 25 to 45% of patients do not respond to this therapy, and more than 40% end up having a relapse [25,48]. Failure of BCG therapy can be defined by several factors, including insufficient therapy, occult invasive or metastatic disease, including progression to MIBC (T2), inadequate immune response, and influence of pre-existing tumor microenvironment [27,48]. Non-responders patients to BCG therapy are at risk of tumor progression, therefore immediate radical cystectomy is recommended to accomplish disease control, when possible [49].

Even with all the advances in diagnosis and specific therapy by staging, disease control has not yet been achieved, due to resistance to BCG immunotherapy and recurrence of chemotherapy-resistant disease [7].

#### 3.2. Immune Checkpoints Inhibitors

ICIs are a monoclonal antibody (mAb) developed specifically to act on negative co-signaling molecules that prevent an effective immune response [50]. The interaction of these inhibitors with the respective immune checkpoints leads to the reinvigoration of tumor activity mediated by T cells, namely, CD8+ T cells [51]. This type of therapy results in an increased local and systemic immune response against tumor cells. Currently, international guidelines include ICIs as second-line therapy after the progression of cisplatin-based chemotherapy [9].

Regarding urothelial carcinoma, monotherapy with ICIs has shown impressive results in clinical trials, both as first and second-line therapies [52]. Nowadays, five ICIs directed to urothelial carcinoma have been approved by FDA: Pembrolizumab, Nivolumab, Avelumab, Atezolizumab, and Durvalumab [9,32,53], as observed in Table 1. Medicines targeting PD-1 are the Nivolumab and Pembrolizumab, against PD-L1 the Durvalumab, Atezolizumab, and Avelumab; and Ipilimumab targeting CTLA-4 [9,54].

**Table 1.** Articles included in the literature review.

| Ref | Author | Year | Type of Study | Drug | Methods | Results |
|---|---|---|---|---|---|---|
| [27] | Annels N et al. | 2020 | R | BCG | Review of the BCG approach and its action on the tumor microenvironment in the NMIBC | Influence of the tumor microenvironment on BCG immunotherapy results; the interaction between cancer, immunity, and BCG |
| [32] | Wołącewicz M et al. | 2020 | R | Pembrolizumab Atezolizumab Nivolumab Durvalumab Avelumab Ipilimumab BCG | Review the current state of immunotherapy | Studies on the different immunotherapies approved by the FDA for BC demonstrate that therapies are well-tolerated and safe therapies |
| [35] | Song D et al. | 2019 | R | Pembrolizumab Atezolizumab Nivolumab Durvalumab Avelumab Ipilimumab BCG | Review the current state of immunotherapy | Intravesical instillation with BCG in the NMIBC and ICIs (anti-PD-1 and anti-PD-L1) in the advanced stage of UC is effective |
| [37] | Massari F et al. | 2018 | R | Combined Therapy | Review the current state of immunotherapy and combined therapies | ICIs have demonstrated a better safety profile compared to chemotherapy Progress studies that investigate the durability, safety, and tolerance of combined therapies and monotherapy in different contexts |
| [39] | Katz H et al. | 2017 | R | Pembrolizumab Atezolizumab Nivolumab Durvalumab Avelumab Ipilimumab | Review the current state of immunotherapy and combined therapies | Main studies with ICIs that led to their approval with UC therapy |
| [42] | Butt S et al. | 2018 | R | Pembrolizumab Atezolizumab Nivolumab Durvalumab Avelumab Ipilimumab BCG | Review the current state of immunotherapy | Studies with ICIs have shown promising activity in the treatment of BC Ongoing studies to investigate the best-tolerated dose in the context of monotherapy and combination therapy |

**Table 1.** *Cont*.

| Ref | Author | Year | Type of Study | Drug | Methods | Results |
|-----|--------|------|---------------|------|---------|---------|
| [43] | Kim H et al. | 2018 | R | Pembrolizumab Atezolizumab Nivolumab Durvalumab Avelumab | Review of the current status of immune checkpoint inhibitors and analysis of the application potential of PD-1 / PD-L1 inhibitors in different contexts | Studies with ICIs have demonstrated lasting long-term responses and tolerable safety profiles Approximately 70 to 80% of patients may not respond to ICIS therapy |
| [44] | Balar A et al. | 2017 | CT | Pembrolizumab | All patients receive 200 mg ofPembrolizumab intravenously every three weeksClinical follow-up every six weeks PD-L1 expression was determined by IHC | After six months, 63% of patients had discontinued therapy Median duration of response has not been reached 42% of patients had disease progression 62% of patients had AE There were 18 deaths associated with therapy |
| [45] | Faiena I et al. | 2018 | R | Durvalumab | Review of Durvalumab therapy | Durvalumab is effective in advanced settings There are still patients who do not respond or show disease progression after initial therapy with Durvalumab Ongoing studies involving Durvalumab in the context of combination therapy |
| [46] | Rouanne M et al. | 2018 | R | Pembrolizumab Atezolizumab Nivolumab Durvalumab Avelumab | Review the current state of immunotherapy | Monotherapy with ICIs has shown good results as second-line therapy Many patients do not respond to monotherapy with anti-PD-1 therapy and anti-PD-L1 therapy Ongoing studies are evaluating ICIs in different contexts, namely, high-risk NMIBC and MIBC, in neoadjuvant and adjuvant treatment or as first-line therapy |
| [47] | Stenehjem D et al. | 2018 | R | Pembrolizumab Atezolizumab Nivolumab Durvalumab Avelumab | Review of the current status of immune checkpoint inhibitors and analysis of the application potential of PD-1/PD-L1 inhibitors in different contexts | The PD-1 and PD-L1 inhibitors have favorable efficacy and safety profiles The only Pembrolizumab demonstrated superiority compared to standard chemotherapy Atezolizumab and Pembrolizumab were well-tolerated in the first-line setting in patients ineligible for cisplatin |
| [48] | Necchi A et al. | 2018 | CT | Pembrolizumab | Patients received 200 mg of Pembrolizumab intravenously every three weeks (three cycles) PD-L1 expression was determined by IHC | 6% of patients discontinued therapy Neoadjuvant therapy with Pembrolizumab was successful in 42% of patients Some AE have been registered |

**Table 1.** *Cont.*

| Ref | Author | Year | Type of Study | Drug | Methods | Results |
|---|---|---|---|---|---|---|
| [49] | Petrylak D et al. | 2018 | CT | Atezolizumab | Nine patients received 1200 mg of Atezolizumab, and 86 patients received 15 mg/kg, both intravenously, every three weeks (16 cycles) PD-L1 expression was determined by IHC | 1% of patients discontinued therapy 67% of patients had AE The therapy administered was well tolerated and provided long-term clinical benefits |
| [50] | Galsky M et al. | 2020 | CT | Combined Therapy | Group A: Atezolizumab with platinum-based chemotherapy Group B: Atezolizumab monotherapy Group C: Placebo with platinum-based chemotherapy In group A, patients received a dose of gemcitabine, carboplatin or cisplatin, two times per cycle In group B, 1200 mg of Atezolizumab per cycle was administered intravenously In group C, one dose per cycle Clinical evaluation every nine weeks | 52%, 54%, and 57% deaths e recorded in group A, B, and C, respectively In group A, 12% of patients had disease progression In group B, 37% of patients had disease progression In group C, 15% of patients had disease progression In all patients, 97% had AE |
| [51] | Suzman D et al. | 2019 | CT | Combined Therapy | Patients received 1200 mg Atezolizumab every three weeks Patients received 200 mg of Pembrolizumab every three weeks PD-L1 expression was determined by IHC | 82% of patients discontinued therapy with Atezolizumab AE associated with Atezolizumab have been reported one death associated with Atezolizumab has been reposted 50% of patients discontinued therapy with Pembrolizumab AE associated with Pembrolizumab has been reported 13 death associated with Atezolizumab have been reposted |

BCG: Bacillus Calmette-Guerin; BC: BC; CT: Clinical trial; AE: Adverse effects; FDA: Food and Drug Administration; IHC: Immunohistochemistry; ICIs: Immune checkpoint inhibitors; MIBC: Muscle Invasive BC; NMIBC: Non-Muscle Invasive BC; PD-1: Programmed cell death-1; PD-L1: Programmed cell death-ligand 1; REF: Reference; R: Review; UC: Urothelial carcinoma.

### 3.3. Anti-PD-L1 Therapies

Atezolizumab is a high-affinity humanized IgG1 mAb that targets PD-L1, inhibiting its bindings to PD-1 [9,48,55,56]. Clinical data argue that Atezolizumab can restore T cell anti-tumor activity in patients with suppressed immunity [48]. It was the first inhibitor of the PD-1/PD-L1 pathway approved by the FDA as a therapy for advanced urothelial carcinoma. Lately, in 2017, it received approval from the European Medicines Agency (EMA) [39,55,56]. Approval was based on the objective response rate (ORR) and average duration of response (DOR). Approved indications for Atezolizumab as therapy for urothelial carcinoma were disease progression during or after platinum-containing chemotherapy and first-line therapy for patients ineligible for cisplatin [55].

Imvigor210 is an open phase II, multicenter study with two cohorts' study. This study aimed to evaluate the activity of Atezolizumab, administered intravenously, with a fixed dose of 1200 mg every three weeks until reaching an unacceptable state of clinical and imaging progression. PD-L1 expression in immune cells (IC) infiltrated in the tumor was evaluated by immunohistochemistry [9]. Patients were categorized by percentage of PD-L1 expression in IC ($IC_0$ 1%, $IC_1$ 1–5%, $IC_{2/3} \geq$ 5%). Disease assessments took place every nine weeks [55]. Involving 199 patients with locally advanced or inoperable metastatic urothelial carcinoma with progression after platinum-based chemotherapy in cohort 1, whilst in cohort 2 were patients receiving Atezolizumab as first-line therapy, due to ineligibility to cisplatin [9,55]. An ORR of 23% occurred, with 9% of patients achieving a complete response. PD-L1 expression in ICS infiltrated in the tumor was assessed. ORR was 9% in the $IC_0$ subgroup, 18% in the $IC_1$ subgroup, and 26% in the $IC_{2/3}$ subgroup. The median overall survival rate (OS) was 15.9 months, while the median progression-free survival (PFS) was 2.7 months. Results have shown that higher PD-L1 expression was associated with higher response rates and longer survival in opposition to what was observed in other therapies in the same scenario, in which PD-L1 expression did not correlate with clinical results [48,55]. The most observed adverse events (AE) were nausea (22%), diarrhea (24%), decreased appetite (24%), and fatigue (52%) [39].

Cohort 2 involved 310 metastatic urothelial carcinoma patients ineligible for cisplatin therapy [55]. In this cohort, ORR was 15%, including a complete response rate with a long extent of response, and 84% of patients were free of progression after 11.7 months of follow-up. Median OS was 11.4 months, while the median PFS was 2.1 months.

The study protocol allowed 120 patients (39%) to receive Atezolizumab after progression with standard chemotherapy, with 17% of them suffering a reduction in tumor volume of at least 30%. PD-L1 expression analysis was also performed by immunohistochemistry as cohort 1 [55]. The prevalence of PD-L1 in IC was significantly increased in basal subtype (60%) compared to luminal subtype (23%). However, no correlation with response rates was observed. Most common AE were constipation (21%), pyrexia (21%), urinary tract infection (22%), nausea (25%), decreased appetite (26%) and fatigue (52%). Severe AE, such as hospitalization, significant disability, or death occurred in 48% of patients [39] (Table 2).

IMvigor211 is a phase III study comparing Atezolizumab therapy with chemotherapy in a second-line after platinum-based chemotherapy in advanced urothelial carcinoma [37]. This study involved 931 patients distributed into two groups: Atezolizumab (*n* = 467) and chemotherapy (*n* = 464). All patients were screened and grouped by PD-L1 expression status ($IC_0$, $IC_1$, and $IC_{2/3}$). A total of 234 patients had a PD-L1 status $IC_{2/3}$, 116 entered in the Atezolizumab group, and 118 the chemotherapy group. Median overall survival (OS) showed no statistically significant differences between the two groups (11.1 months in the Atezolizumab group *vs.* 10.6 months in the chemotherapy group). This study showed that the one-year survival rate was more favorable in the Atezolizumab group than in the chemotherapy group. DOR was of 21.7 months in the Atezolizumab group, while in the chemotherapy group, it was 7.4 months. This study also analyzed the tumor mutation load as a marker for response to therapy in the $IC_{2/3}$ population, concluding that in patients with a higher tumor mutation load, the OS was 1.8 months with Atezolizumab therapy compared to 10.6 months with chemotherapy. This study demonstrated a remission rate

with Atezolizumab in patients with metastatic urothelial carcinoma previously treated with platinum-based chemotherapy treatment. However, Atezolizumab has not shown a beneficial OS compared to chemotherapy in the $IC_{2/3}$ population [55].

Also, patients treated with Atezolizumab during tumor progression showed a better survival rate, compared to patients who discontinued therapy. The average OS was 12.8 months in patients who maintained therapy, while in patients with interrupted therapy, it was 3.6 months [39].

Durvalumab is a humanized IgG1 mAb that blocks PD-1/PD-L1 binding [9,48,55,57]. It was approved by FDA on 1 May 2017, as a second-line therapy for advanced or metastatic urothelial carcinoma [55].

The effectiveness of Durvalumab was studied and analyzed in Study 1108, a phase I/II, multicenter, open, and non-random study [16,55]. This study analyzed anti-tumor activity in a group of 191 patients with advanced or metastatic urothelial carcinoma [9,48]. This study included patients regardless of the status of PD-L1. Quantification of PD-L1 expression in tumor cells was performed, and positivity of PD-L1 expression was defined as PD-L1 ≥ 25% [16]. Patients were treated by intravenous administration of 10 mg/kg of Durvalumab every two weeks for 12 months or until unacceptable toxicity or disease progression. The results show that the subgroup with positive PD-L1 expression shows an ORR of 27.6%, whereas the subgroup with negative PD-L1 expression has an ORR of 5.1%. Patients presented a median PFS and OS of 1.5 and 18.2 months in the total population. In the PD-L1 positive expression, the subgroup has a median PFS of 2.1 months and an OS of 20 months [9,55,57]. Therapy-associated AE has been reported in 63.9% of patients. Fatigue, diarrhea, and decreased appetite were the most common. No grade 4 or 5 AE have been reported, as shown in Table 2 [39]. Durvalumab demonstrated preliminary safety and efficacy in this dose-expansion cohort [55], and currently, continues to be studied as a possible therapy for BC, both as monotherapy and combined therapy [57].

Avelumab is a humanized IgG1 mAb that targets PD-L1 [9,48,55]. It reinforces the immune response against the tumor [48]. It was approved by FDA in 2017, as therapy for patients with metastatic urothelial carcinoma with disease progression during or after platinum-based chemotherapy or patients with disease progression one year after neoadjuvant/adjuvant therapy with platinum-based chemotherapy [48]. This therapy was also approved as second-line [55]. FDA approval was granted based on the JAVELIN study [16,49]. This study involved 249 patients diagnosed with advanced urothelial carcinoma and progresses after first-line chemotherapy, based on cisplatin or patients' ineligibility to cisplatin. The primary objective of this study was the safety of therapy in the dose-escalation cohort of the study and the ORR [9]. Patients were included in the study regardless of PD-L1 status [55]. Prior therapy was administered to 98% of patients, and efficacy assessment was carried out in only 161 patients, who received prior platinum-based therapy and were followed up to six months [54]. ORR was 17%, with a complete response rate of 6% in 161 patients. ORR in patients with tumor expression of PD-L1 ≥ 5% was 24%. Mean DOR was not achieved, although 96% of patients continued to respond to therapy after 24 months. OS was of 6.5 and 8.2 months in the total population and the subgroup with tumor expression of PDL1 ≥ 5%, respectively [48,54,55]. AE were observed in 65.9% of patients, being fatigue (20.5%), asthenia (11.4%) and nausea the most common, as observed in Table 2 [39]. JAVELIN demonstrated the safety and efficacy of Avelumab in BC [55], and subsequently, other studies on Avelumab were initiated, in various therapeutic settings, namely, studies of maintenance monotherapy in patients with metastatic urothelial carcinoma without progression after first-line platinum-based therapy [48].

**Table 2.** Studies on which the Food and Drug Administration relied on therapeutic approval.

| Ref | Agents | Study | Target | Phase | *n* | PD-L1 Expression | PD-L1 Expression (*n*) | ORR% | FDA Approval for UC |
|---|---|---|---|---|---|---|---|---|---|
| [56] | Atezolizumab | Imvigor 210 | PD-L1 | II | 310 | IC ≥ 5%<br>IC ≥ 1%<br>IC < 1% | IC ≥ 5%: 100<br>IC ≥ 1%: 207<br>IC < 1%: 103 | 26%<br>18%<br>8% | 18 May 2016 |
| [56] | Durvalumab | Study 1108 | PD-L1 | I/II | 191 | PD-L1 ≥ 25%<br>PD-L1 < 25% | PD-L1 ≥ 25%: 98<br>PD-L1 < 25%: 79 | 27.6%<br>5.1% | 1 May 2017 |
| [49] | Avelumab | JAVELIN | PD-L1 | Ib | 249 | PD-L1 ≥ 5% | PD-L1 ≥ 5%: 63 | 24% | 9 May 2017 |
| [49] | Nivolumab | CheckMate 275 | PD-1 | II | 265 | PD-L1 > 5%<br>PD-L1 ≥ 1% | PD-L1 > 5%: 81<br>PD-L1 ≥ 1%: 122 | 28.4%<br>23.8% | 5 February 2017 |
| [56] | Pembrolizumab | Keynote-045 | PD-1 | III | 542 | CPS ≥ 10 | Pembrolizumab: 270<br>Chemotherapy: 272 | 21.1%<br>11.4% | 18 May 2017 |

PD-L1: Programmed cell death protein ligand 1; PD-1: Programmed cell death-1; IC: Immune cells; ORR: Objective response rate; CPS: Combined positive score; UC: Urothelial carcinoma.

### 3.4. Anti-PD-1 Therapies

Pembrolizumab is a humanized IgG4 mAb that inhibits the PD-1/PD-L1 interaction.

Pembrolizumab showed anti-tumor activity associated with a satisfactory safety profile [9]. It has been approved for patients with locally advanced or metastatic urothelial carcinoma illegible for chemotherapy (platinum-based), as well as for those patients with disease progression already under platinum-based chemotherapy.

The Pembrolizumab was evaluated in several randomized, open, and active control studies (KEYNOTE-002, KEYNOTE-006, KEYNOTE-010, KEYNOTE-021, KEYNOTE-045) and non-randomized and open studies (KEYNOTE-012, KEYNOTE-052, KEYNOTE-087) [55].

One hundred fifteen patients diagnosed with locally advanced or metastatic urothelial carcinoma were at the beginning involved in KEYNOTE-012 (a non-randomized, open phase Ib study) [48]. PD-L1 expression status in tumor cells was analyzed, and only those with PD-L1 above 1% expression were included in the study, in a total of 33 patients. Patients had a follow-up of 13 months. There were three patients with a complete response and four patients with a partial response. Two months was the average response time, with 10 months of median DOR. Median PFS at 12 months was 15%, median OS at 12 months was 50%. The most observed adverse effects were peripheral edema (12%) and fatigue (18%) [39]. In this study, 26% of PD-L1 positive patients responded to Pembrolizumab therapy [58]

Another study, KEYNOTE-045, a phase III, randomized, two-arm, compared the effectiveness of Pembrolizumab with the effectiveness of chemotherapy chosen by the investigator. This study involved 542 patients who had previously received therapy for metastatic urothelial carcinoma or experienced a recurrence within 12 months of adjuvant or neoadjuvant platinum-based therapy [37]. Every three weeks, 270 patients received 200 mg of Pembrolizumab; while standard second-line single-agent chemotherapy (vinflunine, docetaxel, or paclitaxel) was administered to 272 patients [48,55]. This study aimed to assess OS and PFS in the general population and the positive PD-L1 subgroup. PD-L1 was assessed by immunohistochemistry (9). The average follow-up in both groups was nine months; 28.5% of the patients in the Pembrolizumab group and 33.8% of the patients in the chemotherapy group had CPS with PD-L1 $\geq$ 10%. The results showed an OS of 7.4 months in the chemotherapy group and 10.3 months in the Pembrolizumab group. The ORR was 21.1% in the Pembrolizumab group and 11.4% in the chemotherapy group (56). The PFS was not significantly different. The 12-month survival rate was 43.9% in the Pembrolizumab group and 30.7% in the chemotherapy group (56).

Pembrolizumab as a second-line therapy improved overall survival compared to chemotherapy (10 months versus seven months; $p < 0,01$) in patients with advanced and recurrent urothelial carcinoma [59] (Table 2).

FDA approved Pembrolizumab for therapy in metastatic urothelial carcinoma with disease progression, as well as during or after platinum-based chemotherapy or within twelve months following neoadjuvant treatment with platinum-based chemotherapy. Better tolerance to Pembrolizumab was observed compared to chemotherapy [48,55,58,59].

Nivolumab is a IgG4 monoclonal antibody fully-humanized [9,48,55]. It was approved by FDA for locally advanced or metastatic urothelial carcinoma with disease progression treatment. Being used throughout or after platinum-based chemotherapy or disease progression, as well as in the 12 months after neoadjuvant adjuvant therapy [55].

CheckMate 032, a multicenter, open phase I/II study, included 78 patients with recurrent metastatic urothelial carcinoma after platinum-based chemotherapy, who were treated intravenously with Nivolumab [16,48,55]. The primary objective of this study was ORR [48]. PD-L1 expression in tumor cells was assessed by immunohistochemistry and categorized as positive for expression of PD-L1 $\geq$ 1%. ORR was 24.4% in the total population (78 patients) and 24% in the PDL1 positive subgroup. Median PFS was 2.8 months in the total population and 5.5 months in the PDL1 positive subgroup, and median OS was 9.7 months and 16.2 months, respectively, while median DOR was 9.4 months [55]. AE associated with Nivolumab therapy, namely, grade 3 or 4 toxicity, occurred in 22% of patients; fatigue (3%), maculopapular rash (3%), dyspnoea (3%), lymphopenia (3%), neutropenia

(3%), elevated amylase (4%) and elevated lipase (5%) were the most common. Two patients discontinued treatment because they had pneumonia (grade 4) and thrombocytopenia (grade 4) [39].

In CheckMate 275, a phase II single-arm multicenter study involved 265 patients with advanced urothelial carcinoma progressing after platinum-based chemotherapy. [9]. Patients were administered intravenously with 2 mg/kg of Nivolumab every two weeks as second-line therapy, until unacceptable toxicity was reached or if the disease progresses [9,16]. The primary objective of this study was ORR, as in the CheckMate 032 study. PD-L1 expression on the surface of tumor cells (PD-L1 ≥ 1% or PD-L1 ≥ 5%) was quantified by immunohistochemistry (Dako 28-8) [9,48]. ORR was 19.6% in the total study population, and it was considerably higher in the subgroup with tumor expression of PD-L1 > 5% (28.4%) and lower in the subgroup with tumor expression of PD-L1 ≥ 1% (23.8%) [9]. Median OS was 8.7 months in the general study population, 11.3 months in the subgroup with tumor expression of PD-L1 > 1%, and 5.9 months in the subgroup with tumor expression of PD-L1 < 1%. The safety profile of Nivolumab was similar to the CheckMate 032 study [48]. Median PFS was two months. Median OS was 8.7 months in the general study population and 11.3 months in the subgroup with tumor expression of PDL1 ≥ 1% [55]. AE were observed in 64% of patients, the most common being fatigue (17%). Thirteen patients (5%) had to discontinue therapy, due to unacceptable toxicity [39](Table 2).

This study showed a response to therapy even in patients with low expression of PD-L1, raising the question of whether PD-L1 expression is a reliable predictive marker for response to therapy in BC [48].

### 3.5. Anti-CTLA-4 Therapies

Ipilimumab is a CTLA-4-targeted mAb, currently approved as a therapy for metastatic melanoma [60,61]. In recent years, in several clinical trials, promising results were observed as monotherapy, but also as a combination therapy in various types of tumors. A small study in which Ipilimumab (involving 12 patients receiving two doses of Ipilimumab) was used as neoadjuvant therapy in surgically resectable urothelial carcinoma revealed its potential activity for BC treatment. This therapy was performed before radical cystectomy. In two-thirds of patients, pathological evidence indicating a reduction in tumor staging was observed, while one-third of the patients presented negative urinary cytology.

More recent data suggest that some chemotherapeutic agents mediate the anti-tumor effect by inducing immunogenic cell death. Other therapies, namely, cisplatin, are known to modulate tumor immunity. Combination therapy has been the subject of research for possible use in BC, namely, CTLA-4 inhibitors, inhibitors of immune checkpoints, and chemotherapy. A phase II study of Ipilimumab in combination therapy with chemotherapy of cisplatin and gemcitabine in metastatic urothelial carcinoma involved 36 patients in the absence of therapy who were given six cycles of combination therapy. This study presented an OS of 14.6 months and a survival rate at 12 months of 59%, an ORR of 23%, and a median PFS of eight months. Although the study did not reach an 80% one-year survival rate, it reaffirmed BC.

Immunogenicity [48]. Grade 1 and 2 adverse effects (skin rash, diarrhea, uveitis, and pancreatitis without symptoms) were observed in five of the six patients who received 3 mg/Kg/dose, one patient had ischemic allopathy/optic neuritis.

Patients who have been given 10 mg/kg/dose, four developed grade 1 and 2 AE, including orchitis/epididymitis, elevated transaminases, skin rash, and diarrhea. Three patients showed grade 3 AE with elevated transaminases and diarrhea [39].

Currently, several studies are under way focused on combination therapy of CTLA-4 inhibitors with PD-1 inhibitors [61,62]. Tremelimumab is a humanized IgG2 mAb that targets CTLA-4 [63]. It promotes the activation of cytotoxic T cells in the initial stage of the immune response [64]. Tremelimumab is widely used as therapy in various malignancies, where it works as a combination therapy, usually with Durvalumab [65].

Currently, a study is under way where tremelimumab constitutes a combined therapy with Durvalumab for patients with unresectable BC in stage IV [66]. However, tremelimumab has not yet been approved by the FDA for BC [63,66].

These therapies have different mechanisms of action on different cells of the immune system, as shown in Figure 3.

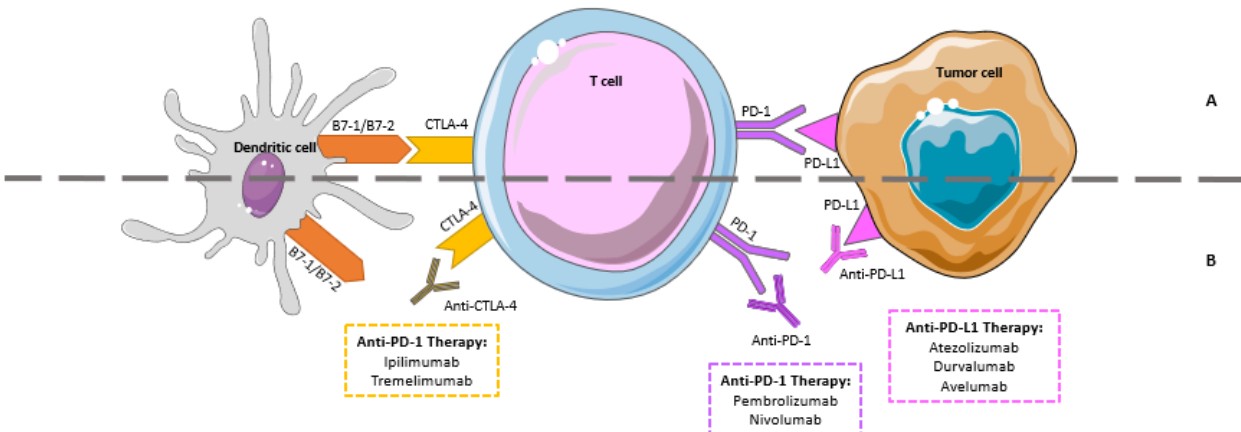

**Figure 3.** Interaction between immune checkpoints and their ligands and the action of immunotherapy with immune checkpoint inhibitors. (**A**) The immune checkpoints, PD-1 and CTLA-4 are expressed in the activated T cells, the binding with the respective ligands on the surface of the tumor cells leads to the inactivation of the T cells, and consequently, prevents the death of the tumor cells; (**B**) Mechanism of action of anti-PD-1, anti-PD-L1, and anti-CTLA-4 immune checkpoint inhibitors. These immune checkpoint inhibitors will inhibit the binding of immune checkpoints to their ligands, ensuring the activation of T cells, favoring anti-tumor activity. PD-1: Programmed cell death-1; PD-L1: Programmed cell death-ligand 1; CTLA-4: Antigen 4 associated with cytotoxic T cells.

Nevertheless, there is a growing body of evidence that a new area emerges, the resistance to ICIs treatment [67]. This resistance can affect the patient treatment outcome, being the patient characterized as (a) individuals that respond initially and maintain the response, identified as responders, (b) individuals that never respond, the innate resistance, and (c) individuals that primarily respond, but eventually progress disease development, the acquired resistance [67–69] At the cellular level the innate and acquired resistance can be the result of insufficient generation of anti-tumor T cells; inadequate function of tumor-specific T cells, or impaired formation of T cell memory [67,70,71].

This area in BC should be deepened, once clarified, which mechanism is involved, might contribute to decreasing resistance to BC ICIs treatment.

## 4. Combined Therapy

After understanding the mechanisms of action of monotherapy compounds, established therapeutic strategies have combined ICIs with other immune therapies, including inhibition of two immune checkpoints with PD-1/PD-L1 and CTLA-4, tyrosine kinase inhibitors, or chemotherapies [50].

A potential synergistic immunotherapeutic activity may be observed by combining PD-1/PD-L1 and CTLA-4 inhibitors therapy. PD-1 and CTLA-4 inhibitor act in different immune response phases (the effector phase or the initial phase, respectively). Therefore, their combination may well potentiate the anti-tumor activity.

Therapy with Durvalumab or in combination with Tremelimumab is being evaluated in different trials, namely: (i) A phase II randomized trial (NCT02527434), where the primary goal is therapy safety; (ii) DANUBE randomized phase III study (NCT02516241), a study in patients with unresectable BC in stage IV. The primary objective of the DANUBE study is PFS, and the results will be assessed according to PD-L1 status and eligibility for cisplatin [57].

There are several studies currently investigating the effectiveness of systemic chemotherapy combined with ICIs in metastatic urothelial carcinoma [66].

IMvigor130 (a multicenter, randomized, placebo-controlled phase 3 trial), compares the administration of Atezolizumab in first-line, with or without platinum-based chemotherapy and placebo chemotherapy in patients with locally advanced or metastatic urothelial carcinoma for its efficacy and safety [72].

KEYNOTE-361 is a phase III, multicenter, randomized, controlled clinical trial, a three-arm study, which analyses Pembrolizumab in combination with platinum-based chemotherapy or monotherapy, compared to standard plus placebo chemotherapy, but there are still no preliminary data. Another ongoing study is the CheckMate-901, a phase III randomized study with four arms, which analyses the combination of Nivolumab and Ipilimumab in the first-line scenario compared to a combination of Nivolumab and standard chemotherapy or chemotherapy alone [9].

In the future, the role of ICIs in different contexts of the disease will continue to be explored, including early, preoperative, and adjuvant stages. Combination of ICIs among themselves and with other agents, namely, immune agents, target drugs, oncolytic viruses, chemotherapy, and radiotherapy, may in the future be the key to success in urothelial carcinoma therapy [62].

## 5. Conclusions

In this review, we sought to expose the current scenario of BC and the new therapies approved for its treatment, highlighting the immune system, tumor microenvironment, and immunotherapies approved by the FDA for BC. Correlation between the immune system and tumor microenvironment was addressed, focusing on the various mechanisms of action on tumor cells. The immune response against cancer results from the balance between activation and inhibition mechanisms of the immune system. Studies have shown that increasing the activation of the immune system is not enough for the patient to be cancer-free, requiring a combination of immunotherapeutic strategies and elimination of immunosuppressive mechanisms.

Previously, BC treatment focused on chemotherapy and surgery, however, with the approval of BCG intravesical therapy and ICIs, immunotherapy gained greater prominence as therapy for BC. In this type of therapy, there is no direct action against cancer, as in other therapies. The basis of this therapy is the activation or stimulation of the immune system, so as not to allow the tumor cells to escape, making it a therapy with greater efficiency and safety. Immunotherapy can be administered alone or in combination with other therapies. There are studies and clinical trials that have evaluated the combination of therapies and shown to be more effective in eliminating tumor cells when compared to monotherapy strategies. We conclude that the correlation between the immune system, the tumor microenvironment, and immunotherapy form a triangle responsible for the improvement of patients' lives. However, there is a need to expand the investigation of immunotherapies in more clinical contexts associated with the different histological subtypes of urothelial carcinoma, as well as the mechanisms of ICIs resistance.

**Author Contributions:** Conceptualization, A.L.S., P.A.-M., D.M., F.M.; methodology, A.L.S., P.A.-M.; investigation, A.L.S., P.A.-M.; data curation, D.M., F.M.; writing—original draft preparation, A.L.S., P.A.-M.; writing—review and editing, D.M., F.M.; supervision, D.M., F.M. All authors have read and agreed to the published version of the manuscript.

**Funding:** This research was funded by the National Funds via Foundation for Science and Technology (FCT), Portugal through Strategic Projects UID/NEU/04539/2019,UIDB/04539/2020 and UIDP/04539/2020 (CIBB).

**Institutional Review Board Statement:** Not applicable.

**Informed Consent Statement:** Not applicable.

**Data Availability Statement:** Not applicable.

**Conflicts of Interest:** The authors declare no conflict of interest. The funders had no role in the design of the study; in the collection, analyses, or interpretation of data; in the writing of the manuscript, or in the decision to publish the results.

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
