# Peer review of "The Impact of Immune Checkpoint-Inhibitors Therapy in Urinary Bladder Cancer"

_onco, doi:10.3390/onco1010002_

Round 1
Reviewer 1 Report
Silva et al. have organized and well written a review on immune checkpoint-inhibitors in bladder cancer using informative figures. Few minor but important changes need to be added before they get accepted. I will highly recommend these suggestions to consider in the revision process.
1) Table 1: rewrite the results column as a summarized conclusion of the study instead of providing only the numbers.
2) Figure 2 is a flow chart, not a prism flow. Please revise it.
3) This is a review article so please you could change 'Material and methods' to Methods (Literature study/survey)
4) Have author included a novel Nanaovaccine-based immunomodulatory approached that have been investigated in several research article.
5) If Nanovaccine based approaches not well-published in Bladder cancer you could improvise future perspectives by mentioning Nanotechnology-based vaccine approaches by referring citing these interesting articles PUBMED: 32776461. You should site as novel examples.
Author Response
Authors: We are delighted that the Reviewers and Editorial Board have found our manuscript meritorious. We are grateful to the Reviewers and their constructive criticism, which have helped us to improve the manuscript.
Please find below a point-by-point reply to the Reviewers’ comments.
Silva et al. have organized and well written a review on immune checkpoint-inhibitors in bladder cancer using informative figures. Few minor but important changes need to be added before they get accepted. I will highly recommend these suggestions to consider in the revision process.
1) Table 1: rewrite the results column as a summarized conclusion of the study instead of providing only the numbers.
R: We thank the Reviewer for bringing this issue to our attention. The results column sometimes is a summarized conclusion and sometimes contains the numbers, according to the type/objective of the original article. The table was revised.
2) Figure 2 is a flow chart, not a prism flow. Please revise it.
R: We thank the Reviewer for bringing this issue to our attention. The alteration was made.
3) This is a review article so please you could change 'Material and methods' to Methods (Literature study/survey)
R: We thank the Reviewer for bringing this issue to our attention. The alteration was made.
4) Have author included a novel Nanaovaccine-based immunomodulatory approached that have been investigated in several research article.
R: The authors think that, although important in the investigation of future treatment options for BC, to approach the theme of nanovaccines was not the aim of this study
5) If Nanovaccine based approaches not well-published in Bladder cancer you could improvise future perspectives by mentioning Nanotechnology-based vaccine approaches by referring citing these interesting articles PUBMED: 32776461. You should site as novel examples.
R: We thank the Reviewer for this comment. The authors think that, although important in the investigation of future treatment options for BC, to approach the theme of nanovaccines was not the aim of this study
Thank you for your valuable commentaries.

Reviewer 2 Report
Relevant topic but excellent reviews exist as benchmark.
Strength of the paper are very recent citations but citations might be inaccurately reproduced or overgeneralized. Scientific content should be carefully revised.
Examples:
P4 l138 TLR not identical with DAMPs but recognize them as receptors (see original citation).
P5 l186 “Inflammatory cells, namely TAMs, facilitate invasion by providing a microenvironment favourable to metastases development”
Poor english usage to a point that the meaning is ditorted. Authors should consider professional language review.
Several examples on the first pages:
P1 l22 consider it
P1 l23 shed light
P1 l34 patients
P3 fig 1 tumor spread to prostate (consider uniform usage of “tumour” or “tumor”)
P3 l119 at the tissue level
P3 l120 The intrinsic
P4 l124 In the extrinsic
P4 l132 for the transition
P4 l134 contribute to
P5 l204-5 “Tumours could present ability to neutralize checkpoints”
Author Response
Authors: We are delighted that the Reviewers and Editorial Board have found our manuscript meritorious. We are grateful to the Reviewers and their constructive criticism, which have helped us to improve the manuscript.
Please find below a point-by-point reply to the Reviewers’ comments.
Relevant topic but excellent reviews exist as benchmark.
Strength of the paper are very recent citations but citations might be inaccurately reproduced or overgeneralized. Scientific content should be carefully revised.
R: We thank the Reviewer for the valuable commentaries.
Examples:
P4 l138 TLR not identical with DAMPs but recognize them as receptors (see original citation).
R: We thank the reviewer for this comment and agree completely. Corrected, sentence re-writed.
P5 l186 “Inflammatory cells, namely TAMs, facilitate invasion by providing a microenvironment favourable to metastases development”
R: We thank the reviewer for this comment and agree completely. The sentence was re-writed and more clear.
Poor english usage to a point that the meaning is ditorted. Authors should consider professional language review.
Several examples on the first pages:
P1 l22 consider it
P1 l23 shed light
P1 l34 patients
P3 fig 1 tumor spread to prostate (consider uniform usage of “tumour” or “tumor”)
P3 l119 at the tissue level
P3 l120 The intrinsic
P4 l124 In the extrinsic
P4 l132 for the transition
P4 l134 contribute to
P5 l204-5 “Tumours could present ability to neutralize checkpoints”
R: We thank the reviewer for this commentary. The English was extensively reviewed.
Thank you for your commentaries.

Reviewer 3 Report
The authors made a comprehensive review of the literature surrounding the approval of immune-checkpoint inhibitors in bladder cancer. The review would be more useful if the following changes are made:
1) in Fig. 2, please include the exclusion criteria used to exclude articles at each stage. Why were so many articles excluded?
2) It will be helpful to compare the anti-PD-1 and anti-PD-L1 therapies among themselves as well as with each other in a table, with respect to ORRs and safety profiles.
3) Potential combination strategies being explored should also be summarized in a table.
4) Resistance to ICIs is an emerging phenomenon. A discussion of resistance mechanisms identified in bladder cancer should be included.
5) A discussion of whether immunotherapies are effective against all types of bladder cancer and the eligibility profiles of patients for immunotherapies will also be helpful.
6) Editing for language clarity and quality is also needed.
Minor comments:
1) Page 2, Lines 79-80: not "invasive muscle" but "muscle-invasive".
2) Page 11, Line 4 and Line 28: not "systematic" but "systemic".
Author Response
Authors: We are delighted that the Reviewers and Editorial Board have found our manuscript meritorious. We are grateful to the Reviewers and their constructive criticism, which have helped us to improve the manuscript.
Please find below a point-by-point reply to the Reviewers’ comments.
The authors made a comprehensive review of the literature surrounding the approval of immune-checkpoint inhibitors in bladder cancer. The review would be more useful if the following changes are made:
1) in Fig. 2, please include the exclusion criteria used to exclude articles at each stage. Why were so many articles excluded?
R: We thank the reviewer for this comment and agree completely. The criteria are now more extensively explained in the text. The flow chart was included to simplify.
2) It will be helpful to compare the anti-PD-1 and anti-PD-L1 therapies among themselves as well as with each other in a table, with respect to ORRs and safety profiles.
R: We thank the reviewer for this comment and agree completely. However, the idea of the article was to review the effectiveness and safety per drug, and not per group. At clinical level, namely in the guidelines, the strength of evidence and the suggestions of therapeutic lines (first-line, second, etc..) is usually given by drug and not by the subgroup of immunotherapy. We think that that “sub-analysis” would be great for a new paper.
3) Potential combination strategies being explored should also be summarized in a table.
R: We thank the reviewer for this comment and agree completely . However, the idea of this paper was to update and give the information of present studied therapies in human clinical trials.
4) Resistance to ICIs is an emerging phenomenon. A discussion of resistance mechanisms identified in bladder cancer should be included.
R: We thank the reviewer for this comment and agree completely. Although it is really important, this topic was not the focus of this paper, but we consider your proposal highly relevant and the authors add to the manuscript some important information regarding ICIs resistance treatment.
5) A discussion of whether immunotherapies are effective against all types of bladder cancer and the eligibility profiles of patients for immunotherapies will also be helpful.
R: We thank the reviewer for this commentary. It was discussed in the introduction and during the text, since the only method available this in the present is the tumor expression % of PD-L1 expression by the tumor. Other evaluation would be interesting, namely if primary tumor and metastasis have the same %, and if that correlates to tumor response. However, this was not the aim of this study.
6) Editing for language clarity and quality is also needed.
R: We thank the reviewer for this comment and completely agree. The English was extensively reviewed.
Minor comments:
1) Page 2, Lines 79-80: not "invasive muscle" but "muscle-invasive".
R: We thank the reviewer for this comment. We performed the alteration.
2) Page 11, Line 4 and Line 28: not "systematic" but "systemic".
R: We thank the reviewer for this comment. Completely agree and the alteration was performed.

Reviewer 4 Report
This paper by Ana Lúcia Silva et al. constitutes an interesting review article about new therapies approved for bladder cancer treatment, highlighting the immune system and tumour microenvironment. The review seems to be very useful and includes a lot of sufficiently detailed and properly ordered information. The paper has been divided into separate sections and for this reason the whole manuscript is clear and readable. The manuscript is generally properly written and the quality of the text is quite good. This is important, especially for review articles, to make sure the manuscript is more easily readable by target audience. Overall, I think the manuscript is worth publishing but some minor points should be address:
- The manuscript should be revised for typewriting, abbreviations, grammar, and style. The most important, example points below:
- line 53, page 11: please correct subscripts in IC indexes
- please correct the dashes in table 1
- please standardize the abbreviations e.g. see lines 149 and 152, page 4
- please use a dot "." as decimal separator, e.g. see line 13, page 14 or line 79, page 15
- please standardize the font, e.g. see lines 236-242, page 6
- Figure 1: the text size is too small. Same for figure 3.
- Figure 3: please explain the abbreviations used.
Author Response
This paper by Ana Lúcia Silva et al. constitutes an interesting review article about new therapies approved for bladder cancer treatment, highlighting the immune system and tumour microenvironment. The review seems to be very useful and includes a lot of sufficiently detailed and properly ordered information. The paper has been divided into separate sections and for this reason the whole manuscript is clear and readable. The manuscript is generally properly written and the quality of the text is quite good. This is important, especially for review articles, to make sure the manuscript is more easily readable by target audience.
R: We are delighted that the Reviewers and Editorial Board have found our manuscript meritorious. We are grateful to the Reviewers and their constructive criticism, which have helped us to improve the manuscript.
Please find below a point-by-point reply to the Reviewers’ comments..
Overall, I think the manuscript is worth publishing but some minor points should be address:
- The manuscript should be revised for typewriting, abbreviations, grammar, and style. The most important, example points below:
R: We thank the reviewer for this comment. It was extensively corrected.
- line 53, page 11: please correct subscripts in IC indexes
R: We thank the reviewer for this comment. We perform the alteration.
- Please correct the dashes in table 1
R: We thank the reviewer for this comment. We remove the dashes.
- please standardize the abbreviations e.g. see lines 149 and 152, page 4
R: We thank the reviewer for this comment. We perform the alteration.
- please use a dot "." as decimal separator, e.g. see line 13, page 14 or line 79, page 15
R: We thank the reviewer for this comment. We perform the alteration.
- please standardize the font, e.g. see lines 236-242, page 6
R: We thank the reviewer for this comment. We perform the alteration.
- Figure 1: the text size is too small. Same for figure 3.
R: We thank the reviewer for this comment. We perform the alteration.
- Figure 3: please explain the abbreviations used.
R: We thank the reviewer for this comment. All the abbreviations were explained in the legend.
Thank you for your commentaries.

Round 2
Reviewer 2 Report
Only selected examples of incorrect english usage have been adressed. Professional language editing service recommended (e.g. https://www.mdpi.com/authors/english).
Minor points:
P5 L189 regulatory Tregs
Table 1 "EA" instead of AE in several rows
P16 L128 "negative urinary cytology for negative cells"
P21 L370 Ref 61 incomplete
Author Response
We are grateful to the Reviewers and their constructive criticism, which have helped us to improve the manuscript.
Please find below a point-by-point reply to the Reviewers’ comments.
Reviewer 2
P5 L189 regulatory Tregs
We thank the Reviewer for bringing this issue to our attention, regulatory was deleted from the Text.
Table 1 "EA" instead of AE in several rows
We thank the reviewer for this comment and completely agree, EA were replaced on the text of Table 1 by AE.
P16 L128 "negative urinary cytology for negative cells"
We thank the reviewer for this comment. The text was replaced by negative urinary cytology.
P21 L370 Ref 61 incomplete
We thank the reviewer for this comment. The Ref was completed:
- Saad and A. Kasi, “Ipilimumab.,”StatPearls". Treasure Island (FL), Jan. 2020, PMID: 32491727.

Reviewer 3 Report
No comments.
Author Response
We are grateful to the Reviewers and their constructive criticism, which have helped us to improve the manuscript.
Please find below a point-by-point reply to the Reviewers’ comments.
Reviewer 3
English language and style are fine/minor spell check required
All document was revised using Grammarly and also by an English native person.
